# Rituximab in Mucous Membrane Pemphigoid: A Monocentric Retrospective Study in 10 Patients with Severe/Refractory Disease

**DOI:** 10.3390/jcm11144102

**Published:** 2022-07-15

**Authors:** Maria Efenesia Baffa, Alberto Corrà, Roberto Maglie, Elena Biancamaria Mariotti, Francesca Montefusco, Carlo Pipitò, Stefano Senatore, Lavinia Quintarelli, Marzia Caproni, Emiliano Antiga

**Affiliations:** 1Department of Health Sciences, Section of Dermatology, University of Florence, 50125 Florence, Italy; alberto.corra@unifi.it (A.C.); roberto.maglie@unifi.it (R.M.); elenabiancamaria.mariotti@unifi.it (E.B.M.); francesca.montefusco@unifi.it (F.M.); carlo.pipito@unifi.it (C.P.); stefano.senatore@unifi.it (S.S.); emiliano.antiga@unifi.it (E.A.); 2Rare Skin Diseases Unit, Azienda USL Toscana Centro, Department of Health Sciences, University of Florence, 50125 Florence, Italy; lavinia.quintarelli@unifi.it (L.Q.); marzia.caproni@unifi.it (M.C.)

**Keywords:** mucous membrane pemphigoid, autoimmune bullous skin disease, rituximab

## Abstract

Rituximab (RTX) is a monoclonal antibody directed against CD20 antigen indicated in an increasing number of immune-mediated diseases. While its efficacy in pemphigus vulgaris has been widely investigated, only a few data about its possible role in pemphigoid diseases have been reported in the literature. Accordingly, herein we evaluated a case series of patients with mucous membrane pemphigoid (MMP) treated with RTX. We included patients with a history of severe/refractory MMP who received at least one cycle of intravenous RTX between May 2018 and December 2021 and had 6 months of follow-up time. Disease control (DC) was our early endpoint, while complete remission (CR) and partial remission (PR) were late endpoints. CR off-therapy, relapses, and adverse events were evaluated as well. Our population included 10 MMP patients. Eight out of ten patients (80%) achieved DC in a mean of 8 weeks, while two patients with ocular MMP were non-responders. Among the eight patients who achieved DC, two reached CR off therapy, two CR on minimal therapy, and two achieved PR on minimal therapy. In our case series, the addition of RTX to conventional therapies was demonstrated to be safe and effective in reaching rapid disease control in the majority of refractory MMP patients.

## 1. Introduction

Mucous membrane pemphigoid (MMP) includes a group of rare autoimmune blistering diseases with a predominant mucosal involvement, characterized by the presence of IgG and/or IgA autoantibodies against the chorio-epithelial junction [1].

MMP is a heterogeneous disease regarding both disease localization and severity. In some affected areas, e.g., the conjunctiva or the upper aerodigestive tract, MMP lesions tend to resolve with scarring and can lead to important sequalae, respectively partial or total vision loss and life-threatening obstructions; on the other side, the fibrotic potential of MMP lesions in other anatomical sites, e.g., the oral mucosa, is less relevant and lesions can heal without scarring [2]. The treatment of MMP is challenging due to multiple reasons, including the older age of affected patients, variable comorbidities that at the same time result from and limit the use of immunosuppressive treatments, and the absence of effective therapies for preventing or reversing scarring [3].

Over the recent years, several case reports and case series highlighted rituximab, a monoclonal antibody targeting CD20, as a potentially effective therapeutic option for recalcitrant/refractory MMP. Based on accumulating evidence, in 2021, the European Guidelines on diagnosis and management of mucous membrane pemphigoid recommended RTX, either alone or in combination with conventional therapies, as a second-line treatment in severe MMP and as a third-line treatment in mild/moderate disease [4].

Here, we report our institutional experience with the use of RTX in patients with MMP.

## 2. Materials and Methods

In the present study, we included all patients with a history of severe/refractory MMP who received at least one cycle of intravenous RTX between May 2018 and December 2021 at the Department of Health Science, Section of Dermatology, University of Florence, and had at least 6 months follow-up time after the first RTX administration.

MMP diagnosis was made by evaluating clinical features together with either the evidence of linear deposits of IgG, IgA, or C3 alongside the basement membrane zone (BMZ) at the direct immunofluorescence microscopy (DIF) and/or indirect immunofluorescence (IIF) performed on salt-split skin (SSS) [5].

For each patient, we considered the following characteristics: sex, age, clinical presentation, oral steroid dosage, number of precedent adjuvant therapies, and months between diagnosis and first RTX infusion.

Patients received two different protocols: intravenous RTX 1000 gr twice (two weeks apart) according to the high-dose RA protocol or four weekly infusions of RTX 375 mg/m^2^ as per the lymphoma protocol [6]. Additional 1–2 cycles were administered in case of non-adequate response or relapse. B-cell count and infectious disease screening were performed for all patients before each RTX cycle.

Patient records were evaluated separately by three of the authors (MEB, EBM, and AC).

We took disease control (DC) as the early endpoint, complete remission (CR) and partial remission (PR) as the late endpoint according to MMP definitions by Murrell et al. [7]. For ocular MMP, where it is very difficult to assess both early and late endpoints, we considered, respectively, resolution of the erythema and absence of new ocular lesions to assess DC and the absence of active ocular inflammation for >2 months, as proposed by You et al. to assess remission [8]. Relapses, adverse events, and oral corticosteroid tapering were evaluated as well.

Descriptive statistical methods were used to analyze the data.

## 3. Results

The patient population included ten MMP patients, five females and five males. Patients’ characteristics are summarized in Table 1. Six patients had oral MMP with no ocular involvement, and five of them had a multisite involvement. The remaining four patients had an ocular monosite MMP.

Eight out of ten patients had a history of refractory disease due to inefficacy/intolerance of immunosuppressive or adjuvant therapies. Only two patients received RTX as a first-line drug due to the rapid and severe onset of the disease: patient 6 presented with extensive erosions of the oral cavity, larynx, and pharynx, unresponsive to high doses of oral prednisone, and patient 8 with very severe ocular involvement.

The mean dosage of prednisone before RTX administration was 26.7 mg. The mean age at the time of the first RTX infusion was 68.5 years (range 52–90). The mean time between the diagnosis and the first cycle of Rituximab was 14.6 months (range 2–36).

All patients received RTX without discontinuation of their pre-existing therapeutic regimens, mainly consisting of prednisone, dapsone, azathioprine, methotrexate, and mycophenolate.

Disease control (DS) was reached in 8 out of our 10 patients (80%) in a mean time of 8 weeks (range 5–14). All six MMP patients without ocular involvement reached DC. The remaining two patients did not reach this early endpoint and were considered non-responders.

Among the eight patients who achieved DC, two reached CR off therapy, two CR on minimal therapy, and two achieved PR on minimal therapy, while the remaining two patients having monosite ocular MMP were not able to gain any further improvement after DC.

Five out of eight patients (62%) experimented with relapse in a mean of 5 months (range 4–9), requiring an additional cycle of RTX. Patient 4 had a second relapse after 46 weeks, requiring the third cycle of RTX. The relapse reported by patient 2 was reasonably induced by the anti-SARS-CoV2 vaccine due to the strict temporal relationship (only a few days as reported by the patient) between the first dose of BNT162b2 and the worsening of her clinical condition.

The median dose of oral prednisone decreased by 7.74 mg after 3 months from the first RTX cycle. After 6 months, two patients were able to cease oral prednisone, while patient 1 increased daily intake due to a relapse.

Early adverse events were fatigue and augmented sweating. There were no major late adverse events; the reported ones consisted of asthenia, diarrhea, cephalea, hyperglycemia, dyspnea, and temporary lip paresthesias. No relevant infectious events were reported.

Main results are summarized in Table 2. 

## 4. Discussion

Rituximab is a murine/human monoclonal antibody that specifically targets CD20, a B-cell surface antigen, leading to a rapid depletion of the circulating CD20+ B-cell population [9,10]. Originally developed for the treatment of non-Hodgkin lymphomas, RTX has been recently approved for rheumatoid arthritis, granulomatosis with polyangiitis, microscopic polyangiitis, and pemphigus vulgaris (PV) as well [11,12]. While more than 100 studies, including a prospective multicenter controlled trial [13], assess the efficacy of RTX in PV, the research about RTX usage in MMP, also given the rarity and the heterogeneity of the condition, has been considerably slower and only a small number of articles explored the topic.

In one study, MMP was reported to have a mean interval of 14.5 weeks to reach DC after RTX treatment [14], which can be considered a slower response compared to pemphigus, in which DC is usually reached within 2 months [15,16]. This observation might be explained considering that, in pemphigoid diseases, pathogenetic factors other than autoantibodies play a relevant role in the development of the lesions [17,18]. By contrast, our patients attained DC after a mean interval of 8 weeks after the first RTX administration; the latency before DC was shorter than in other studies, possibly due to the continuation of the other therapies such as methotrexate, azathioprine, and mycophenolate during RTX cycles.

Regarding ocular MMP, patients 6 and 7, who were able to attain DC, had a considerably shorter latency time between diagnosis and the first RTX administration compared to the other two ocular MMP patients, who were considered non-responders. This may suggest that early administration of RTX during the disease course may allow achieving a better outcome.

Overall, the response rate in our case series was 60%, with 40% being able to reach a CR. Considering only the oral MMP group, 100% of patients were able to gain a late endpoint (CR or PR), while, as expected, patients with monosite ocular MMP had worse outcomes. One of the non-responders ocular MMP was even diagnosed with conjunctival intraepithelial neoplasia during the follow-up period, requiring six applications of interferon-alpha-2b. In our case series, the overall response rate was not high as in other studies. For example, La Roux-Villet et al. reported an overall response rate of 92% after the second cycle of RTX [19], while Tovanabutra et al. observed 76% of CR in their population [20]. Moreover, a recent systematic review reported 112 MMP cases treated with RTX showing better outcomes than those of our study, with an overall response of 94% and CR in 70% of the patients. [21] These differences can be explained by the inclusion of single case reports that can be affected by publication bias and the fact that in more than 60% of the cases MMP subtype was not reported. Most importantly, while we used standardized criteria to assess MMP disease activity [7], Lytvyn et al. used different outcome measures, probably depending on the heterogeneity of the studies included in their systematic review [21].

We had a 62% of relapsing MMP patients in a mean of 5 months (range 4–9). Our results are consistent with current literature: four different papers reported relapses between 45% and 100% after a mean of 9 months (range 4–15.2). Relapse is frequent among MMP patients treated with RTX, which does not seem to grant a stable remission [14,19,20,22,23].

Given the small sample size, it was not possible to evaluate if there were any differences in the response or in the relapse rate between the high-dose AR protocol and the Lymphoma protocol. In the literature, there is also no clear evidence of the superiority of one protocol over the other. Of interest, You et al., in a recent study, use the Foster protocol in the treatment of ocular MMP, suggesting that a low maintenance dose of RTX could be particularly beneficial in the ocular disease [8].

In our case series, RTX was well tolerated, as no major adverse events arose. Moreover, there were no relevant infectious episodes among our patients, although the increased risk of infection is still considered a major safety concern [21,24].

Although this study has some limitations, mainly consisting of the retrospective nature and small sample size, our observations were overall consistent with the current literature, adding further evidence in favor of the use of RTX in the therapy of MMP.

In conclusion, our data suggest that RTX might be a good choice in recalcitrant/refractory oral MMP, in conformity with the most recent European guidelines [4]. On the other hand, we believe that further studies are required to explore its role in ocular MMP, where a more tempestive treatment with RTX might lead to a rapid stabilization of the disease.

## Figures and Tables

**Table 1 jcm-11-04102-t001:** Patient characteristics.

PatientNo.	Age	Sex	Diagnosis-First RTX Cycle Time (Months)	Number of Failed Therapies	Systemic Therapy at RTX Administration	Localization (Onset)
1	63	F	14	3	Prednisone, Azathioprine	Genital (onset), oral
2	56	F	12	3	Prednisone, Dapsone	Oral (onset), genital, cutaneous
3	59	F	24	4	Prednisone, Methotrexate	Oral (onset), nasal, genital, cutaneous
4	69	F	4	3	Prednisone, Dapsone, Azathioprine	Oral (onset)
5	52	F	36	3	Prednisone, Dapsone	Oral (onset), cutaneous
6	90	M	3	N/A	Prednisone	Oral (onset), pharyngeal, laryngeal
7	74	M	8	2	Dapsone, Mycofenolato mofetil	Ocular
8	75	M	2	1	Deflazacort	Ocular
9	83	M	24	N/A	Dapsone, Mycofenolato mofetil	Ocular
10	64	M	10	2	Dapsone, Mycofenolato mofetil	Ocular

Abbreviations: RTX—rituximab.

**Table 2 jcm-11-04102-t002:** Therapeutic outcomes in mucous membrane pemphigoid patients treated with rituximab.

Patient No.	Diagnosis	Weeks to DC after First Cycle	Response	No. of Weeks to Response	Follow-Up, Months	Relapse	Months to Relapse after First Cycle	Adverse Events
1	Oral MMP	6	CRMT	40	25	YES	6	Immediate: fatigue, augmented sweating, hyperglycemia Late: asthenia
2	Oral MMP	4	PRMT	14	12	YES	6.5	Diarrhea
3	Oral MMP	7	CROT	29	19	YES	9	Lip paresthesia, cephalea, epigastralgia
4	Oral MMP	9	CRMT	71	36	YES	4	
5	Oral MMP	7	CROT	69	25	NO	--	Dyspnea
6	Oral MMP	6	PRMT	16	9	NO	--	--
7	Ocular MMP	12	--	--	33	YES	8	--
8	Ocular MMP	14	--	--	6	NO	--	Fever
9	Ocular MMP	--	N/A	--	12	N/A	N/A	--
10	Ocular MMP	--	N/A		23	N/A	N/A	--

Abbreviations: DC—disease control; MMP—mucous membrane pemphigoid; CRMT—complete remission on minimal therapy; CROT—complete remission off minimal therapy; PRMT—partial remission on minimal therapy.

## Data Availability

Not applicable.

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
