# Peer review of "Rituximab in Mucous Membrane Pemphigoid: A Monocentric Retrospective Study in 10 Patients with Severe/Refractory Disease"

_jcm, 2022, doi:10.3390/jcm11144102_

Round 1

Reviewer 1 Report

In this retrospective study, the authors report their experience with Rituximab in 10 patients with severe or recalcitrant mucous membrane pemphigoid. The authors correctly address that, in contrast to pemphigus vulgaris, more evidence and data is needed regarding treatment of MMP with Rituximab. 

The report is adequate and the data is well presented, however, the report might improve with minor additions. 

1. Because of the study design and its limitations of a small retrospective study, I would suggest to add the number of patients in the title. For example 'Rituximab in mucous membrane pemphigoid; a monocentric retrospective study in 10 severe or recalcitrant patients'.

2. The patients received two different protocols of RTX; please address these differences in the results and discussion. Please also mention in the discussion the RTX protocols used in the referred studies, of which the response rates are compared. 

3. The authors mention the B-cells are measured before RTX. This is of significant value to report the B-cell respons during or after treatment. If the data is available, please add this to the results.

4. The outcomes of monosite ocular MMP can be very difficult to assess. What is disease control or partial remission in a patient with ocular scarring? Please address this in the discussion. 

Minor comment:

- Please check drug names in English in text and table

Author Response

Comments and Suggestions for Authors

“The report is adequate, and the data is well presented, however, the report might improve with minor additions.”

 Response: we thank the reviewer for the positive consideration of the manuscript.

  1. Because of the study design and its limitations of a small retrospective study, I would suggest to add the number of patients in the title. For example 'Rituximab in mucous membrane pemphigoid; a monocentric retrospective study in 10 severe or recalcitrant patients'.

Response: we thank the reviewer for his/her comment. We modified the title as suggested.

  1. The patients received two different protocols of RTX; please address these differences in the results and discussion. Please also mention in the discussion the RTX protocols used in the referred studies, of which the response rates are compared. 

Response: we thank the reviewer for his/her comment. Given the small sample size, it was not possible to evaluate if there were any difference in the response or in the relapse rate between the high-dose AR protocol and the Lymphoma protocol.

  1. The authors mention the B-cells are measured before RTX. This is of significant value to report the B-cell respons during or after treatment. If the data is available, please add this to the results.

Response: we thank the reviewer for his/her comment. We didn’t include the exact value of B-cell count because post-treatment data are not available for all the patients.

  1. The outcomes of monosite ocular MMP can be very difficult to assess. What is disease control or partial remission in a patient with ocular scarring? Please address this in the discussion. 

Response: we thank the reviewer for his/her comment. We indicated the assessment methods in the “material and method” section.

Minor comment:

- Please check drug names in English in text and table

Response: we apologies for the mistakes. We corrected all the drugs names in the tables and in the text.

Reviewer 2 Report

This was a single-center retrospective case series of 10 patients treated with rituximab for mucous membrane pemphigoid. The conclusions are reasonable. However, I'm not sure what new information is added by this case series. A recent systematic review (Lytvyn 2021 JAMA Derm) found 112 patients reported in the literature who were treated with rituximab for MMP. The metaanalysis shows rates of remission similar to that seen here.

In general, citations are not very complete. This does not cite the metaanalysis or even some of the larger case series of rituximab for MMP (for example, You 2017 among others).

Author Response

Referee: 2
The conclusions are reasonable. However, I'm not sure what new information is added by this case series. A recent systematic review (Lytvyn 2021 JAMA Derm) found 112 patients reported in the literature who were treated with rituximab for MMP. The metaanalysis shows rates of remission similar to that seen here.

Response: In our opinion, together with adding the experience of a tertiary center, our results are relevant since they highlight the benefits of a early initiation of RTX therapy in ocular MMP and confirm that RTX might be a good choice in recalcitrant/refractory oral MMP, in conformity with the most recent European guidelines, as we wrote in the conclusions of our paper.  Regarding the second comment, although we cited in the text the most relevant studied included in the metanalysis by Lytvyn et al., we agree with the comment of reviewer and cited the latter in the discussion.

In general, citations are not very complete. This does not cite the metanalysis or even some of the larger case series of rituximab for MMP (for example, You 2017 among others).

Response: we thank the reviewer for his/her comment.  As suggested, we included the article by You at al. in our report.

Round 2

Reviewer 2 Report

Thank you for adding references. I still think that this small case series needs to be better compared to the wider experience of MMP treated with rituximab, in your discussion. I suggest you spend a few sentences specifying how your response rate and relapse rate and subsets of disease compare with the aggregate results of the 112 patients compared in the systematic review.

Author Response

Referee: 2
Thank you for adding references. I still think that this small case series needs to be better compared to the wider experience of MMP treated with rituximab, in your discussion. I suggest you spend a few sentences specifying how your response rate and relapse rate and subsets of disease compare with the aggregate results of the 112 patients compared in the systematic review.

Response: we thank the reviewer for his/her comment. As suggested, we compared our results with those of the systematic review by Lytvyn et al., highlighting the main differences and the possible reasons.
